# Modality-Agnostic Topology Aware Localization

Farhad G. Zanjani          Ilia Karmanov          Hanno Ackermann          Daniel Dijkman

Simone Merlin          Max Welling          Fatih Porikli

Qualcomm AI Research*

{fzanjani,ikarmano,hackerma,ddijkman,smerlin,mwelling,fporikli}@qti.qualcomm.com

## Abstract

This work presents a data-driven approach for the indoor localization of an observer on a 2D topological map of the environment. State-of-the-art techniques may yield accurate estimates only when they are tailor-made for a specific data modality like camera-based system that prevents their applicability to broader domains. Here, we establish a modality-agnostic framework (called *OT-Isomap*) and formulate the localization problem in the context of parametric manifold learning while leveraging optimal transportation. This framework allows jointly learning a low-dimensional embedding as well as correspondences with a topological map. We examine the generalizability of the proposed algorithm by applying it to data from diverse modalities such as image sequences and radio frequency signals. The experimental results demonstrate decimeter-level accuracy for localization using different sensory inputs.

## 1   Introduction

Self-localization or localizing objects are primary tasks in navigation and surveillance systems. This problem has been the subject of many studies in the machine learning community. It has been addressed in several areas such as visual odometry [Engel et al., 2017, Brahmbhatt et al., 2018], visual simultaneous mapping and localization (VSLAM) [Davison et al., 2007, Mur-Artal et al., 2015], self-localization [Arth et al., 2011, Sattler et al., 2019, Sarlin et al., 2021], etc., and many approaches have been proposed.

Recent localization methods achieve decimeter precision in the positioning of a moving camera in an indoor environment by leveraging the advances in neural networks [Brahmbhatt et al., 2018, Kendall et al., 2015]. However, such techniques are highly entangled with the modality of data in use; thus, applying such specialized algorithms to other modalities is often not possible. For example, the existing visual odometry and VSLAM techniques that rely on visual features or camera projection models are incompatible with different sensory systems like radio frequency (RF) or audio signals. Still, all instantiate the same problem. In contrast to existing solutions tailored for a particular modality, we formulate the localization problem in terms of low-dimensional manifold learning to represent the input samples in their intrinsic space and transport them to a given topological map by inferring correspondences between intrinsic space and map. This approach allows a broader and more generalizable solution that can be used with data from different modalities.

---

*Qualcomm AI Research is an initiative of Qualcomm Technologies, Inc.

35th Conference on Neural Information Processing Systems (NeurIPS 2021).

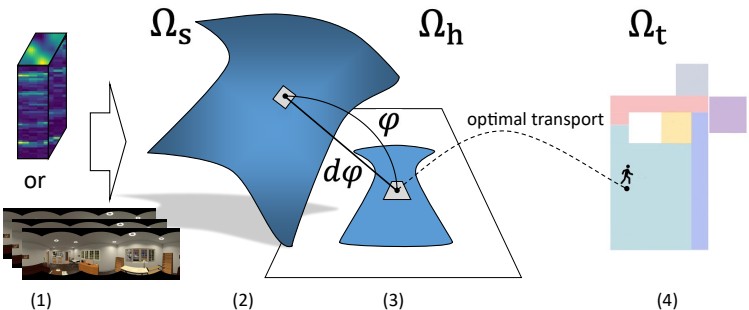

Figure 1: Schematic of the proposed modality-agnostic localization method; (1) different input modalities, for instance, radio frequency (RF) signals or video sequences; (2) the input signals are assumed to change gradually, thus are on a smooth manifold in input space $\Omega_s$; (3) the positions are encoded in an intrinsic space $\Omega_h$. Please refer to Sec. 3.3 for details about the mapping; (4) correspondences between $\Omega_h$ and the positions on the floormap is estimated by means of optimal transportation. Please refer to Sec. 3.2 for details.

Here, by localization, we aim to pinpoint the location of a single observer at each time instance on a topological map by analyzing the collected sensory data. Unlike VSLAM, our method does not build a map; we assume that it is given as a prior. Nevertheless, the topological map can have different forms. For example, it can be an approximate sketch of the 2D floor plan or a precise 2D CAD draft. While the observer moves in the environment and visits different locations, the measured signal such as video frames, depth images, or WiFi *channel state information* (CSI) encode the 2D location of the observer as well as the geometry of the space and other modality-specific factors, depending on the type of the sensory system in its high-dimensional ambient space. However, the *intrinsic* manifold that encodes the positional information (coordinates) lies only either in a 2D or 3D space (3D in case the altitude of the observer varies). Hence, a nonlinear dimensionality-reduction method that maps the input samples into their intrinsic 2D/3D embedding can yield a solution to the localization problem. Incorporating domain knowledge (e.g. camera projection models for visual sensing or wave propagation equations for RF sensing) may reduce the task of finding the mapping to a parametric regression problem; however, the obtained solution would again be modality-specific. Instead, we build on our intuition that the transformation from high-dimensional input to the intrinsic space can be modeled as a parametric mapping, learnable through a neural network between data manifold and the 2D coordinates.

In addition, we observe that the temporal data only changes gradually as the observer moves; for instance, consecutive images in a video stream are similar. This observation implies the data reside in a smooth, locally connected manifold. We further postulate that an embedded latent manifold of much lower dimensionality represents the locations of the observer in the environment. Based on this, we revisit the localization problem and reformulate it as a learning task of the embedding space of data. Since we are given the room/zone labels and a topological map yet not the correspondences between the embedding space and the map, we adopt optimal transportation (OT) to estimate them.

The proposed method does not rely on modality-specific priors, instead it builds on correlation between the observer location and the measured signal. Since we make no assumption on the transformation, the proposed method is applicable to a large spectrum of different sensor modalities. Fig. 1 illustrates the proposed approach.

## 2   Related Work

**Localization** has been the subject of many studies in robotics, computer vision, and RF sensing. As such, classical VSLAM methods [Davison et al., 2007, Mur-Artal et al., 2015] mainly build a 3D point cloud map of the scene while estimating the motion of the camera in the video data. Modern VSLAM approaches are usually composed of several carefully engineered modules, such as tracking, mapping, re-localization, loop closure, graph optimization, key-frame selection, or post-processing by bundle adjustment [Henriques and Vedaldi, 2018]. Many of these modules make strong assumptions on the

visual features or a camera projection model. Even recent efforts to replace some of these modules with learnable architectures borrow those assumptions in their methodology [Chen et al., 2020] or the method requires a strong supervision like known position and orientation [Kendall et al., 2015], full 3D models [Brachmann and Rother, 2018, Sarlin et al., 2021], or image pairs [Toker et al., 2021]. Employing visual features and camera projection models can also be seen in visual odometry [Engel et al., 2017]. While these advances report decimeter-level accuracy in visual localization, they are not applicable to other sensory systems such as RF or audio signals. Their success demands major engineering efforts in transferring similar ideas to a new data modality.

**Manifold learning** is being used extensively in fields like computer vision, natural language processing, and data mining. In manifold learning, the data points are assumed to lie on a smooth manifold $\mathcal{M} \subset \mathbb{R}^n$ in $n$-dimensional measured ambient space, while they are sampled from a distribution, near or on lower-dimensional sub-manifold $\mathcal{N} \subset \mathbb{R}^s$, where $n > s$. The minimum number of variables needed to describe such a distribution is known as the intrinsic dimension. The task of manifold learning is to find a smooth map $\Phi : \mathcal{M} \to \mathcal{N}$. If the data has intrinsic dimensionality of $m$, according to the Whitney Embedding Theorem [Lee, 2013], we know that $\mathcal{M}$ can be embedded smoothly into $s = 2m$, using a homeomorphism $\Phi$. However, it is often impossible to obtain an isometric embedding directly in the intrinsic space [McQueen et al., 2016]. As far as the smooth embedding preserves the topology of $\mathcal{M}$, this might be sufficient for many dimension reduction purposes but when preserving the geometry of embedding is desirable, finding an isometric embedding is required. Several previous works in non-linear dimension reduction [Tenenbaum et al., 2000, Verma, 2012, Weinberger and Saul, 2006, McQueen et al., 2016] have been driven by the desire to find a smooth embedding of low dimension that is isometric. Since almost all embedding algorithms suffer from distortion, there are several works that propose a different technique for mitigating this issue by filtering out the border samples [Rosman et al., 2010], introducing a Riemannian relaxation [McQueen et al., 2016], or setting some constraints on the embedding [Agrawal et al., 2021]. Depending on the amount of imposed distortion in the intrinsic embedding, the performance of a downstream task that is applied on the representation of the sample in the embedding space would degrade. In localization problems, when there is no ground truth positions available, mapping from a distorted intrinsic space into the target map can be difficult and prone to a high error.

**Optimal transportation** has received considerable interest from the topology community [Peyré et al., 2019]. When the data is associated with geometrical properties, optimal transport metrics (also called Wasserstein distance or Earth Mover distance) measure the spatial variations between probability distributions of source and target domains. Correspondence matching is one of the successful applications of optimal transport [Mémoli, 2011, Peyré et al., 2016, Solomon et al., 2016]. Given a transport cost function, the Wasserstein distance computes the optimal transportation plan between two measures. Recent progress on efficient computation of the optimal transport by introducing entropy regularization and Sinkhorn's matrix scaling algorithms reduced the computational cost of optimal transport several orders of magnitude than of the original transport solver [Cuturi, 2013]. As shown in [Genevay et al., 2018], computing the OT loss and its gradient can be tractable by using Sinkhorn fixed point iterations.

Finding the transformation for representing the data points on a given topological 2D map requires knowing a set of correspondences in the localization problem. In case the correspondences are available, learning a transformation is straightforward. However, in an unsupervised setting, estimating such a transformation can be very difficult.

In what we propose, the correspondences (i.e. coupling matrix) between the 2D embedding and the target topological map is learned by using the OT algorithm. We find a coupling matrix (also called transport plan) representing the correspondences between two domains by minimizing the cost of transportation. In the next section, we explain how the transportation cost is parameterized by the representation of a neural network in the intrinsic embedding of data.

## 3 Embedding learning and transportation to the intrinsic space

In this work, we formulate the localization problem in the context of manifold learning and optimal transportation. Our contributions are as follows:

1. We propose a model that jointly learns the intrinsic embedding and its transportation into a topological map of the environment in a weakly-supervised style by using gradient descent

optimization. Such a joint optimization mitigates the distortion of the intrinsic embedding as the model constrains it to resemble the topology of the target map.

2. The proposed localization method does not make any assumptions about the data modality in use. In that sense, it is modality-agnostic and can be applied to a large family of sensory systems. From the system setup point of view, our method can extend to both active and passive positioning problems.

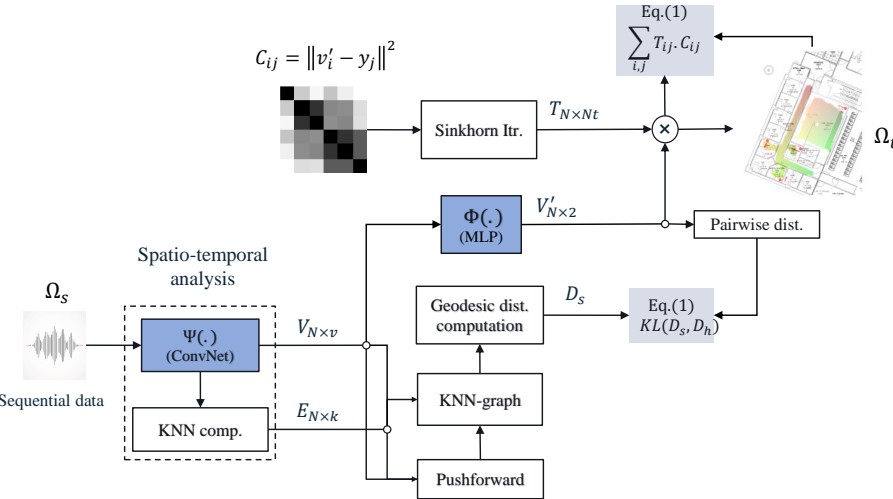

Figure 2: Block diagram of the method. A set of prototype vectors and their KNN are computed by spatio-temporal analysis of the sequential input data. A KNN graph is constructed by using the prototype vectors as its nodes (V) and the KNN indices (E) as the edges (i.e. connectivities). By estimating the pushforward at the location of each node on the data manifold, the small distances between neighboring nodes in the 2D intrinsic space are approximated. The distance between non-adjacent nodes is approximated by shortest path through the graph. The 2D intrinsic representation is learned by preserving the pairwise distances and jointly finding the correspondences with a given target floor plan ($\Omega_t$).

## 3.1 Problem and theoretical motivations

Let $\Omega_s \in \mathbb{R}^n$ be the ambient space of some measured signal and $\Omega_h \in \mathbb{R}^m$ with $m \subset \{2,3\}$ be its intrinsic space. We are interested in finding the representations in $\Omega_h$ of the discrete samples $X_s = \{x_i^s\}_{i=1}^{N_s} \in \Omega_s$. We assume the measured samples $X_s$ lie on a smooth (Riemannian) manifold in $\Omega_s$; thus, the manifold is locally connected. This assumption intuitively holds since the data is a temporal sequence that does not change much between two consecutive instances in time. We also assume that the topological map $\Omega_t \in \mathbb{R}^m$ that represents the geometry of the environment is known, yet no correspondences between these two spaces are available. This map can be in the form of a 2D sketch or a floor plan of a building. Localizing the observer in $\Omega_t$ requires finding a mapping between $\Omega_s$ and $\Omega_t$.

Using manifold learning techniques, one can estimate the embedding in $\mathbb{R}^m$ and consequently seek a transformation to map the embedding into $\Omega_t$. When we do not have access to correspondences between the embedding and target domains, finding an isometric embedding that preserves the pairwise distances between samples is desirable as the transformation to the target would be less complex. Hence, methods like Isomap [Tenenbaum et al., 2000] which preserve the local and global distances, have an advantage. However, due to the non-convexity of $\Omega_s$, it is often impossible to obtain an isometric embedding directly in the intrinsic space [McQueen et al., 2016]. Since walls usually partition interior spaces, there may not be a direct path between any two arbitrary points on the floor plan. Therefore, due to the non-convexity of the sample set, usually, the embedding suffers from a severe distortion that a simple isometric transformation is not sufficient for alignment with the target domain. As an alternative, we propose finding a map between the input and the intrinsic spaces that both preserves the pairwise distances between samples and aligns $\Omega_h$ and $\Omega_t$.

We formulate the localization problem as follows. Consider $\Phi : \Omega_s \rightarrow \Omega_h$ be a smooth map between input and intrinsic embedding. The map $\Phi$ can be implemented by a neural network e.g. an MLP. We define $(D_s, D_h) \in \mathbb{R}^{N_s \times N_s}$ as distance matrices between samples in $\Omega_s$ and $\Omega_h$, respectively. So the entries of $D_h$ can be computed like $d_{ij}^2 = \|\Phi(x_i) - \Phi(x_j)\|^2$. For now, let's assume we have access to the geodesic distance-matrix $D_s$, which contains all pairwise geodesic distances of $X_s$ on the input manifold. Later, in section 3.3, we explain how $D_s$ can be approximated. It is worth mentioning that training the $\Phi$ by minimizing the $\|D_s - D_h\|^2$ and using gradient descent optimization leads to a parametric approximation of the MDS algorithm [Pai et al., 2019]. However, this formulation is ill-posed when $X_s$ is a non-convex set since comparing the geodesic distances with Euclidean distances by using $\|\cdot\|^2$ is only valid inside a convex region.

For solving the correspondence problem, the Gromov-Wasserstein discrepancy between two distance matrices was used [Peyré et al., 2016]. There, the coupling between two sets of distances is found by a regularized optimal transport. In this work, our proposed model learns *both* the map $\Phi$ *and* simultaneously by estimating a coupling matrix ($T \in \mathbb{R}^{N_s \times N_t}$). Training the model therefore amounts to minimizing the loss

$$\min_{\Phi, T} L(D_s, D_h) + \mu \sum_{ij} T_{ij} \cdot C_{ij} \quad \text{with} \quad C_{ij} = \|\Phi(x_i) - y_j\|^2, \tag{1}$$

where the first term $L(.,.)$ is a dissimilarity measure between two distance matrices and the second term is the Sinkhorn distance between the samples in the embedding and the target topological map. The scalar $\mu$ is a contribution weight. The $C \in \mathbb{R}^{Ns \times Nt}$ and $T \in \mathbb{R}^{Ns \times Nt}$ are the cost of transportation and the coupling matrix. Here, $Y = \{y_j\}_{j=1}^{N_t} \in \Omega_t$ is defined by dense sampling from the target domain. The embedding representation can have any arbitrary rotation and scale with respect to their true positions on the floor plan. Here, by using zone/room labels, we constrain the transportation to seek for a valid assignment. To do so, we penalize the transportation when the zone labels of input $x_i$ and the target $y_i$ are different. This can be done simply by multiplying the cost $C_{ij}$ by a constant factor $\alpha_{ij} > 1$ when the labels are different and $\alpha_{ij} = 1$ if they have a same label. Since the distance matrices, $D_s$ and $D_h$, are in two different domains, instead of using $L = \|\cdot\|^2$, We propose to use the Kullback-Leibler divergence (KL) for $L$ [Cuturi, 2013]. Fig. 2 shows the block diagram of the proposed model.

## 3.2 Regularized Transport with Differentiable Sinkhorn Distance

One major advantage of regularizing the optimal transport problem is that it becomes solvable efficiently using Sinkhorn's algorithm [Genevay et al., 2018]. In computing the entropy constraint of the Sinkhorn distance as proposed in [Cuturi, 2013], we are looking for a coupling matrix $T$ that satisfies

$$T(C, p, q) = \underset{T \in \gamma(p,q)}{\operatorname{argmin}} \langle T, C \rangle - \frac{1}{\lambda} H(T). \tag{2}$$

Here, $p$ and $q$ are probability distributions of samples in source $\Omega_s$ and target $\Omega_t$ domains and $\gamma(p, q)$ is their joint probability. The $C \in \mathbb{R}^{Ns \times Nt}$ is cost matrix for transporting mass between two domains. In Eq. (2), the term $H(T) = -\sum T \log(T)$ is the entropy of coupling $T$. The solution of (2) is [Cuturi, 2013]:

$$T(C, p, q) = \operatorname{diag}(a) K \operatorname{diag}(b), \tag{3}$$

where $K = e^{-\lambda C} \in \mathbb{R}_+^{Ns \times Nt}$ is the so-called Gibbs kernel associated to $C$ and $(a, b) \in \mathbb{R}_+^{N_s} \times \mathbb{R}_+^{N_t}$ can be computed using the Sinkhorn-Knopp iterative algorithm:

$$a \leftarrow \frac{p}{Kb} \quad \text{and} \quad b \leftarrow \frac{q}{K^\top a}, \tag{4}$$

where $\top$ denotes the transpose of a matrix and the division is element-wise. When there is no prior knowledge about the location of the observer in the environment, a uniform distribution can be assigned to $p$ and $q$. As proposed in [Genevay et al., 2018], using the Sinkhorn iteration is an efficient way for differentiating Eq. (2). By running Eq. (4) for $M$ iteration, and putting the obtained $(a, b)$ in Eq. (3), we can compute the derivative $\frac{\partial T}{\partial K}$ and consequently $\frac{\partial T}{\partial C}$. Hence, Eq. (1) is differentiable. As the cost matrix $C$ depends on $\Phi$, the gradient can be back-propagated to optimize the parameters of

the neural network $\Phi$. The pseudo code in Algorithm (1) describes the procedure of jointly learning the embedding and the transportation.

The value of the regularization coefficient ($\frac{1}{\lambda}$) in Eq. (2) controls the distribution of $T$. A high entropy on the coupling matrix $T$ means a uniform distribution over its entries that makes it a smooth assignment matrix. In contrast, a low entropy means the $T$ makes a hard assignment. Regularizing the Eq. (2) with high entropy makes the optimization stable [Genevay et al., 2018], at the cost of a higher error in finding the correct correspondences. When training the model, we apply a linear annealing schedule to the weight of the entropy regularization from an initial high value to a final small value. This range of $\lambda$ is a hyper-parameter of the model and is fixed in all our experiments.

---

**Algorithm 1:** Jointly learning the embedding and the transportation plan.

---

**Input** : $X_s = \{x_i\}_{i=1}^{N_s} \in \Omega_s$, $Y = \{y_i\}_{i=1}^{N_t} \in \Omega_t$, and $D_s \in \mathbb{R}^{N_s \times N_s}$
**Output :** model parameters $\Phi$ and coupling matrix $T \in \mathbb{R}^{N_s \times N_t}$
Initialize: $\Phi$, $T$, and $\lambda = \{\lambda^{(0)}, ..., \lambda^{(max)}\}$
**while** $t < Max_{itr}$ **do**

$\quad \hat{D}_{ij} = \|\Phi(x_i) - \Phi(x_j)\|^2$;
$\quad C_{ij} = \alpha_{ij} \cdot \|\Phi(x_i) - y_j\|^2$ ; $\qquad$ // constants $\alpha_{ij} \geq 1$
$\quad T = Sinkhorn(C, \lambda^{(t)})$ ; $\qquad\qquad$ // Eq. (3) and Eq. (4)
$\quad \mathcal{L} = D_{KL}(D_s \| \hat{D}) + \mu \cdot \langle T, C \rangle$ ; $\quad$ // Eq. (1)
$\quad \Phi^{(t+1)} \leftarrow \Phi^{(t)} + \eta \frac{\partial \mathcal{L}}{\partial \Phi}$ ; $\qquad\qquad$ // $\eta$ is learning rate

**end**

---

### 3.3 Computing the geodesic distance-matrix

The objective function in Eq. (1), requires pre-computing the distance matrix $D_s$ that represents the pair-wise distances in the training set $X_s$. Since the inputs $X_s$ are distributed on a non-convex manifold in input space, the pairwise geodesic distances between non-neighboring samples should be measured. For an efficient computation of the geodesic distances, three steps are taken: (1) Reducing the size of $X_s$ by finding a set of representative samples (also called prototype vectors or landmarks) and computing their k-nearest neighbors; (2) estimating the push-forward metric for estimating the Euclidean distances between neighboring prototypes in the embedding; (3) estimating the pairwise geodesic distances between non-neighbor prototypes, using a shortest path algorithm; In the following, we explain each step.

#### 3.3.1 Finding a set of prototypes and their KNN

Similar to landmark Isomap [De Silva and Tenenbaum, 2003], instead of computing all pairwise distances in a large training set of $X_s$, only the pairwise distances in a set of $N_s$ representative samples – called prototype vectors – are computed. Finding a set of prototypes by using farthest-point sampling [Pai et al., 2019] and consequently measuring the Euclidean distance between them in the high-dimensional input space have some drawbacks. Even for the 2D manifolds in $\mathbb{R}^3$, such as surfaces with holes or self-intersection, finding the KNN, by using Euclidean distance is susceptible to short-circuiting on the manifold.

Instead, we propose to learn a metric space for computing both the prototype vectors and their neighbor indices by training a neural network on the triplet sampled data [Hadsell et al., 2006]. Each triplet set contains two samples that are temporally close, and a third one that is distant. We apply an upper bound to the maximal temporal distance. The network learns to produce similar feature vectors for samples, based on their temporal vicinity by minimizing its triplet margin loss:

$$\mathcal{L}(x_i^a, x_i^p, x_i^n) = max\Big(0, d(h_i^a, h_i^p) - d(h_i^a, h_i^n) + \alpha\Big) \quad \text{where} \quad h_i = \Psi(x_i). \tag{5}$$

In Eq.(5), the symbol $\Psi$ denotes the function of the neural network, $d$ measures the Euclidean distances, and $(h_i^a, h_i^p, h_i^n) \in \mathbb{R}^\nu$, $\nu < n$, are the output vectors of the network, produced from the $i$th set of anchor ($x_i^a$), positive ($x_i^p$) and negative ($x_i^n$) instances and the scalar $\alpha$ is a constant margin.

After convergence, the data samples ($X_s$) are partitioned into $N_s$ clusters by applying a standard k-means clustering on the obtained features set $h$. As prototypes, we use the centroid vectors of each

class. Furthermore, by measuring the pairwise Euclidean distances between the feature set, the K nearest neighbors of each prototype are found.

### 3.3.2 Approximating the distances in embedding

For estimating the small distances between neighbor samples in the embedding, measuring the Euclidean distance in the input space is not informative [Dsilva et al., 2015]. For more reliable estimation, the unimportant sources of variability need to be suppressed. As in [Dsilva et al., 2015], using the Mahalanobis distance is representative of distance along the local principal direction of data manifold and can be used as an approximation of small Euclidean distances in the embedding:

$$\|\Phi(x_i) - \Phi(x_j)\|^2 \approx \frac{1}{2}[x_j - x_i]^\top \cdot [C^\dagger(x_i) + C^\dagger(x_j)] \cdot [x_j - x_i], \tag{6}$$

where $C(x_i)$ is the measured local covariance matrix of data at the location of sample $x_i$, and $\dagger$ denotes the Moore-Penrose pseudoinverse. For computing Eq. (6), we estimate the covariance matrix of samples in each cluster. By using Eq. (6), the pairwise distances between neighboring prototype vectors in the embedding space can be approximated. When the distances are small enough, such an estimation is similar to the *pushforward* in differential geometry that is an approximation to the differential of map ($\partial\Phi$) between tangent spaces of two manifolds.

After computing all pairwise distances between each prototype and its KNN, we create a KNN-graph similar to the procedure in Isomap [De Silva and Tenenbaum, 2003] and the distances between non-neighboring samples are estimated by using the Dijkstra's shortest path algorithm [Dijkstra et al., 1959]. Then, the geodesic matrix $D_s$ in the embedding space is known and Eq. (1) can be evaluated for training the model.

## 4 Experiments

We applied the proposed model on three different data modalities. For details about the implementation, hyper-parameters and training please refer to the supplementary material.

### 4.1 Synthetic data - 2D maze environment

To validate our approach without the added complexities of data, we first setup an experiment with a simple 2D Maze environment[1] [Henriques and Vedaldi, 2018, Parisotto and Salakhutdinov, 2017]. In this simulation, an agent walks through a maze from a starting point to the end point. The starting and end points are chosen randomly on the outer walls of the maze (see Fig.3). The solution path between these two points is considered as a trajectory. We create many trajectories by randomly choosing several different starting and end points. A square window, centered at the location of agent is used for sampling an image patch from the environment. A collection of sampled patches is used as the training set. The model predicts the position of observer with $0.7$ pixel error for the maze environment of size $24 \times 24$ pixels. More inference results for number of trajectories are shown in the supplementary (see Fig. 2).

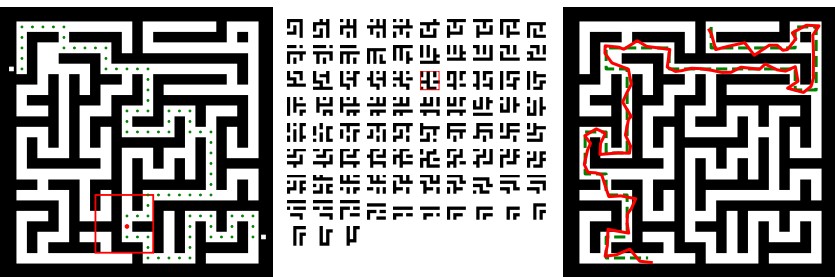

Figure 3: Synthetic maze environment of size $24 \times 24$ pixels; (left) An example of a path and the sampling window; (middle) extracted patches for the shown path; (right) an example of the ground truth (in green) and the predicted (in red) trajectories

---

[1]https://github.com/theJollySin/mazelib (GNU General Public License v3.0)

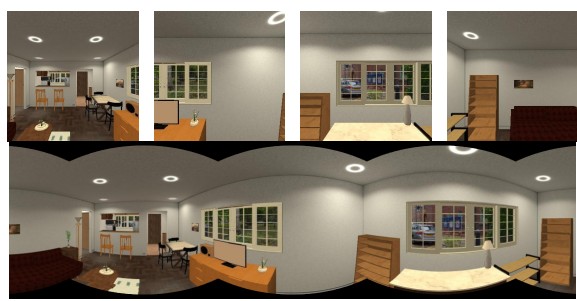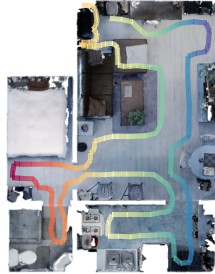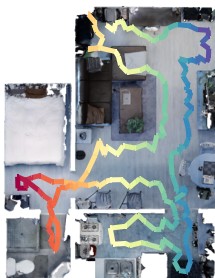

Figure 4: An example of the iGibson environment; (left) the quadruple-view at the location of observer and the constructed panorama image; (middle) a ground truth trajectory; (right) prediction

Table 1: First row: indexes of the 15 scenes of iGibson (*Rs*, *Merom-0*, *Merom-1*, *Pomaria-0*, *Pomaria-1*, *Pomaria-2*, *Ihlen-0*, *Ihlen-1*, *Benevolence-0*, *Benevolence-1*, *Benevolence-2*, *Beechwood-0*, *Beechwood-1*, *Wainscott-0*, *Wainscott-1*). Second row: Localization errors (in meters).

| | Environments | | | | | | | | | | | | | | |
|---|---|---|---|---|---|---|---|---|---|---|---|---|---|---|---|
| index | 1 | 2 | 3 | 4 | 5 | 6 | 7 | 8 | 9 | 10 | 11 | 12 | 13 | 14 | 15 |
| error | 0.62 | 1.05 | 0.73 | 0.64 | 0.69 | 0.83 | 1.02 | 1.03 | 0.73 | 0.62 | 0.74 | 0.71 | 0.98 | 1.04 | 1.14 |

## 4.2 Camera-based localization

In case of using the image modality for localization by learning the data manifold, independent of camera orientation, the input data requires to contain 360 degree image sequences of several indoor spaces. To the best of our knowledge, such a dataset is not publicly available. However, we can take advantage of existing indoor 3D scan dataset and by navigating a camera in the 3D scans create videos. With this aim, we setup an experiment by using the *iGibson* dataset [Shen et al., 2020]. The dataset consists of indoor 3D scans of 15 buildings that can be explored interactively [Xia et al., 2020]. The scans are created from real homes and offices. The iGibson allows us to create a set of trajectories in each environment that simulates the navigation of a robot agent, equipped with a camera. We create several image sequences by navigating the camera through all rooms/zones of each environments. We leave out some of the image sequences for the test set. For sampling from the data manifold, at each location of agent, we capture four orthogonal-view images by rotating the camera with multiple of $90°$ with the up-direction being the rotation axis. These four images are used for constructing a panorama image.

The panorama sequences and the bird-eye view of floor plan are the inputs to the model. Visiting the same location in the environment through several trajectories with a different azimuth angles of the camera introduces a column-wise circular shift in the panorama images. To make the spatio-temporal analysis of data to be invariant with respect to this rolling effect, we normalize the panorama images. We know that cyclic rotations of panorama images are equivalent to an addition of linear phase in the Fourier domain [Pajdla and Hlavac, 1999, Sturzl and Mallot, 2006]. Thus, by computing the FFT of each panorama image and then estimating and removing the linear phase in the horizontal direction and transforming it back into image domain, we normalize each panorama image. Fig. 4 shows an example of four images, taken at the location of observer, a ground-truth trajectory, and its prediction. Table 1 reports the localization errors for different environments in the dataset. More qualitative results have been shown in the supplementary.

## 4.3 Passive WiFi localization

Localization of a moving target in the pervasive WiFi signal has received some attention of the machine learning community [Fan et al., 2020]. Positioning through WiFi can be useful in the domain of home, enterprise and industrial automation and robotics. It is not affected by poor light conditions and can thus work in the dark, and it can work across walls. Also, a device without a video-camera is more likely to be adopted by a user since it largely mitigates issues of privacy related to video images. Our experimental setup is designed to mimic a real-world deployment and thus we only use three receivers and one transmitter for a large multi-room environment where most areas lack

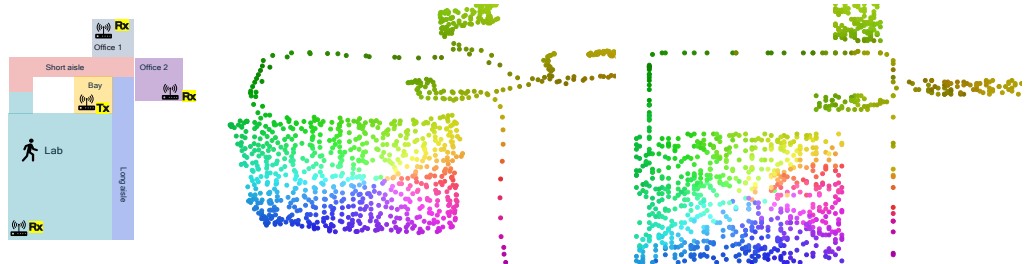

Figure 5: WiFi localization; (left) floor plan topological map; (middle) ground-truth positions of observer; (right) inference. The points color encodes the x-y locations of person for the ground-truth and their correspondences with the prediction.

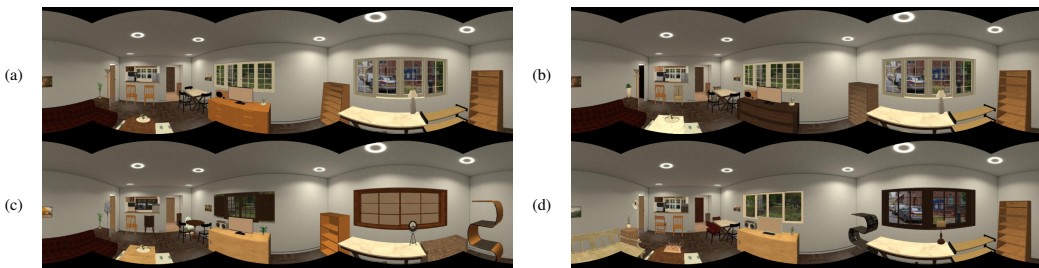

Figure 6: Example of perturbation in the environment; (a) shows a reference environment that has been used for the training; (b-d) show panoramas which correspond to 70%, 50% and 30% of the objects identical with those in (a).

line-of-sight. We consider *passive* positioning, i.e. the only source of information are the reflections of the electromagnetic waves from the body of the moving target. We collected a data set by using four commercial IEEE 802.11 access points (AP), operating in the 5GHz band, and deployed them in the test environment of size $14 \times 20$ meters in a building. The environment contains three rooms, two long aisles and a large lab (see Fig. 5). We use three receivers with eight antennas each, and a transmitter with a single antenna. Each receiver collects the *Channel State Information* (CSI) at periodic intervals. The CSI represents the channel between the transmitter antenna and each of its 8 receiver antennas, across 208 frequency tones that span the transmission bandwidth. Hence, the CSI is represented as a multidimensional tensor of complex numbers of dimension 8x1x208 per each WiFi packet. Here, we only use the magnitude of CSI signal. We recorded about five minutes of CSI data from all three receivers, while a person freely walks through different locations in the environment. For ground-truth positions, we used several 3D cameras, installed in the environment. The algorithm achieved an average error of 1.2 meters for locating the person on the 2D floor plan image of the environment. Fig. 5 visualizes the predicted locations of the person in the building.

### 4.4 Ablation Studies

We did an ablation study on the robustness of the camera-based localization by altering the visual features and object instances in the environment. This has been done by randomly exchanging both texture and appearance of objects in the scene. Fig. 6 shows several examples of such variations with respect to the original environment that has been used for training. By measuring the accuracy the algorithm achieves on the altered scenes in a validation run, we are able to determine how susceptible the algorithm is when it is tested on the perturbed images. To estimate mean and standard deviation, we repeated this test ten times with ten differently randomized scenes. The results can be seen in Table 2.

Table 2: Mean and standard deviation of localization error (cm) under different amount of changes in the environment.

| Rate | Amount of perturbation on the test set | | | | | |
|---|---|---|---|---|---|---|
| | 0% | 10% | 20% | 30% | 50% | 70% |
| Error (mean, std) | 61.9 | $61.87 \pm 0.09$ | $62.2 \pm 0.09$ | $62.82 \pm 0.17$ | $63.77 \pm 0.14$ | $64.07 \pm 0.26$ |

## 5 Discussion

The localization problem includes a large group of systems and methods that by using a sensory system and a computational algorithm aims to find the location of observer on a given topological map. Depending on the type of sensory system, several approaches have been presented in the literature. However, the existing solutions, even the data-driven methods, mostly work for a specific modality. This is mainly because these algorithms are built based on some strong assumptions about the characteristic of input data like image features or sensory system like the projection model of the camera. These assumptions can easily be violated if the algorithm applies on a different data modality like RF signals or vice versa. In this work, we formulated the localization problem in the context of manifold learning by finding the 2D intrinsic space of data that leads to a more generic solution for a large family of localization problems. Finding an isometric embedding for the realistic noisy data under limited amount of samples, is highly erroneous and the computed embedding usually suffers from a severe distortion. Such a distortion imposes difficulty in finding a precise transformation to the topological map in an weakly-supervised style. In this work, we proposed to learn the embedding and the transformation to the target topological map jointly by gradient descent optimization. Since the exact position of target on the topological map is unknown for the model, we employed optimal transport for finding the correspondence (i.e. coupling) between the intrinsic embedding and the target geometry. Our experiments on both synthetic and realistic data with different modalities show the effectiveness of the proposed method.

## Broader Impact

In a positive prospect, we believe that our model contributes to further the development of less domain-specific framework for the indoor localization problem and can inspire future research works. This can drive many applications of localization in other less developed domains than vision-based systems.

One of the limitations of our proposed method is that we assume the main source of temporal variations in measured signal is due to the displacement of a single agent/person in the environment. However, this assumption can be violated when the motion of other objects/people in the environment causes a large variation in the signal. Such uninformative variations with respect to the location of the observer can deteriorate finding the correct KNN samples on the data manifold. While minimizing the triplet loss, a high dependency between the spatial and temporal information of the observer was assumed. This issue may be less important for the camera-based localization and in general for an active localization problem (i.e. when the observer carries a device that is being used for measurements like RF signal of a cell phone or a camera). In passive localization cases, like localization of the person in WiFi medium by the reflected signal from the surface of the body, motions of other people cause a catastrophic impact on the manifold learning as then the measured signal contains a superposition of more than one sample on the data manifold.

Depending on the data modality in use, the measured signal can be almost identical for two different locations in the environment. For example, the high visual similarity of two rooms of a hotel can easily cause mistakes for many image-based localization algorithms. The high visual similarity can even confuse a human. Although this may not be the case for WiFi localization, it can suffer as well when the geometry of the reflectors in the environment be very symmetrical with respect to a virtual line between the transmitter and a receiver.

## Acknowledgments and Disclosure of Funding

This work was funded by Qualcomm Technologies, Inc.

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
