# Modality-Agnostic Topology Aware Localization
## – Supplemental Material –

Farhad G. Zanjani     Ilia Karmanov     Hanno Ackermann     Daniel Dijkman

Simone Merlin     Max Welling     Fatih Porikli

Qualcomm AI Research[*]

## 1 Discussion on the choice of hyper-parameters

**Triplet sampling** was implemented based on the temporal vicinity of samples. Since the input is sequential, for each sample (called anchor) in the sequence, we consider a small and a large temporal window with predefined fixed widths. These two temporal windows are centered at the timestamp of the anchor. Any sample inside the smaller temporal window can be considered as a positive sample and any sample outside the small window but inside the large window can be considered as a negative sample. The widths of the temporal windows roughly depend on the speed of the observer in the environment. The idea is that the negative samples need to include data at which the observer has sufficient displacement from its previous location (i.e. anchor sample). The upper bound of the width of the larger temporal window is set as large as possible. However, if it is chosen too large, the observer could revisit the same location. In other words, the observer returns to the location where the anchor sample was collected and this would be degenerative for learning the spatial neighborhood.

**The number of clusters** ($N_s$) is a trade off between spatial accuracy and computational cost. A larger number of clusters leads to a dense sampling from the data manifold but it requires training the model on a larger set to learn the embedding and to find the transportation cost to the target domain.

**The neighborhood size** ($K$) should be large enough to guarantee the connectivity in the KNN-graph but too large a value degrades the expressiveness of KNN-graph and the accuracy of its computed geodesic distances.

## 2 2D-maze experiment

We generated a 2D maze of size $24 \times 24$ pixels by using the *Mazelib* library[1]. By randomly selecting the two ends on a path and by using the solvers that are provided by the library, a path through the maze between these two points is found. We generated 100 trajectories by randomly choosing 100 pairs of endpoints on the outer walls of the maze. The agent observes the environment through an image patch of size $6 \times 6$ pixels, centered at its location.

For learning the prototype vectors and their KNN samples, a 3-layer MLP network (of 36-128-64-6 units per layer) with ReLU activations is trained by using the Adam optimizer with a learning rate of $\eta = 1\mathrm{e}{-3}$. After convergence, k-means clustered the output feature of the network into 200 clusters. By using Euclidean distance, KNN samples (K=3) of each prototype vectors are found. As explained in Section (3.3.2), the pushforward at the center of each cluster is estimated and the Euclidean distances in the embedding space are approximated. By using Dijkstra's algorithm, all non-neighbor

---

[*]Qualcomm AI Research is an initiative of Qualcomm Technologies, Inc.

[1]https://github.com/theJollySin/mazelib (GNU General Public License v3.0).

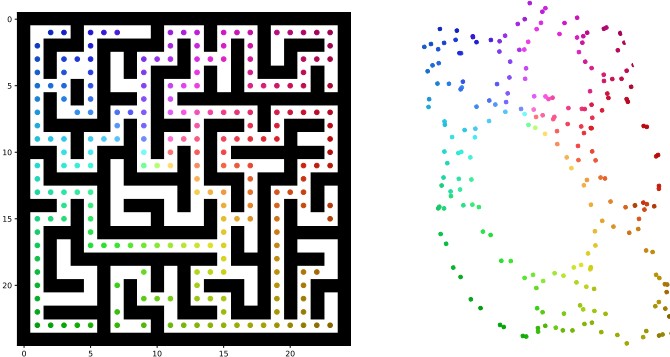

Figure 1: Visualization of (left) a number of trajectories of agent in the 2D maze and (right) their 2D embedding. The colors encode x-y positions in the maze.

distances are estimated and the matrix $D_s$ is determined. Fig. 1 shows the 2D embedding of the $D_s$ for some trajectories.

For learning the map $\Phi$, a 3-layer MLP network (of 36-128-64-2 units per layer) with ReLU activation functions was used. We trained the model for 3000 epochs with the Adam optimizer and a learning rate of $\eta = 1e-4$. The weight of entropy equalizer ($\frac{1}{\lambda}$ in Eq.(2)) is changed from 0.1 to 0.01 linearly during training and the Sinkhorn algorithm runs for 100 iterations per epoch. Fig. 2 shows several ground-truth (in green) trajectories and their predictions (in red) in the maze.

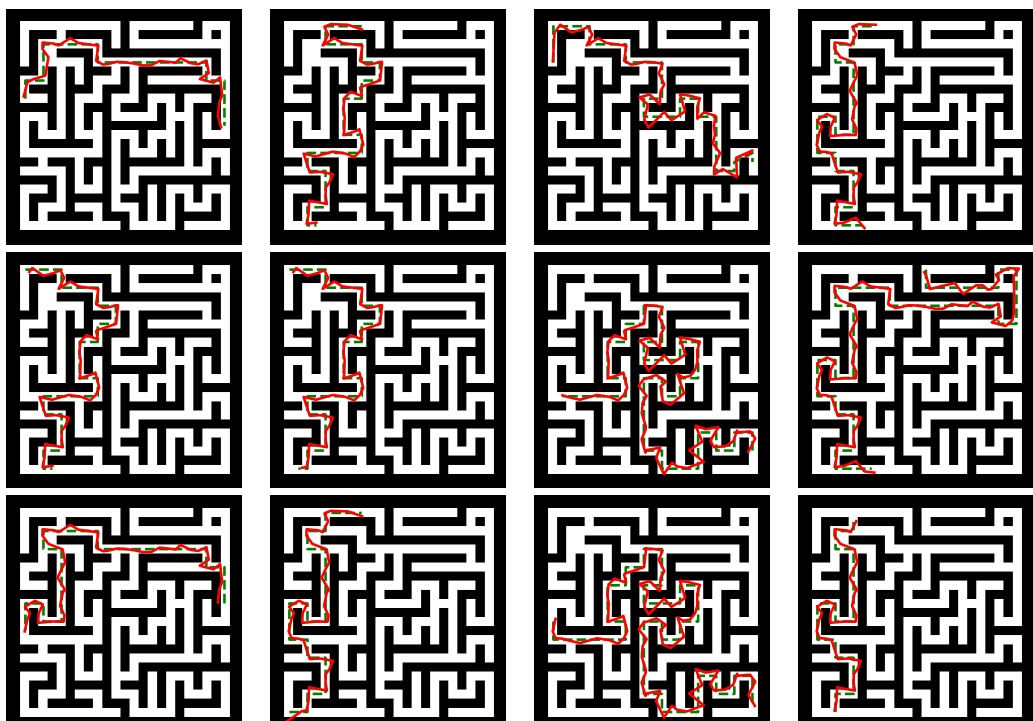

Figure 2: Inference results for the maze experiment; observer locations (in green) and their predictions (in red).

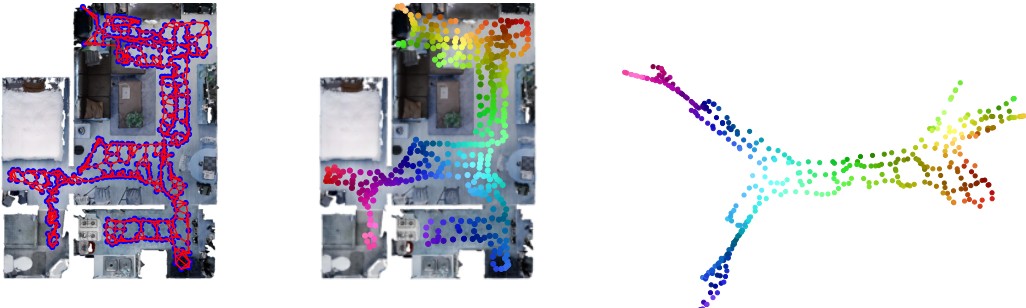

Figure 3: Visualization of (left) the locations of prototype vectors (cluster centroids) and the constructed KNN graph, (middle) ground truth locations, and (right) 2D representation of the images in the embedding space.

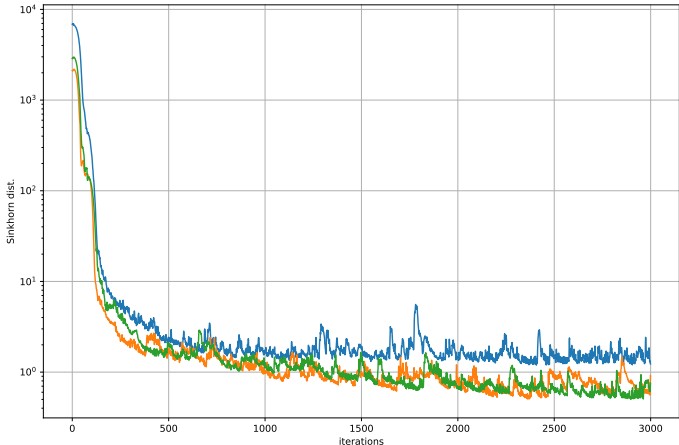

Figure 4: The training Sinkhorn distance for three different iGibson environments that shows the stability of the optimization.

## 3 Camera-based localization experiment

All 15 iGibson environments are used in the experiment. A robot agent traverses in each environment. At each location, it captures four images by rotating the camera in multiples of $90°$ with the up-direction being the rotation axis. Each image is generated with $200 \times 200$ pixels. By stitching together these four images, a panorama image is generated. On average, 2000 sequential images are used for training and the same amounts are left out for the test. We standardized all panorama images by using zero-phase normalization as explained in the paper. We trained a 3-layer MLP for finding the prototype vectors and their KNNs. The input to the network is the (120-dimensional) PCA feature of each panorama image. After convergence, by using the k-means algorithm, 400 clusters and their KNNs (k=3) were found. Fig. 3 shows the constructed KNN-graph and the 2D embedding of the person locations by processing the images for an environment.

For learning the map $\Phi$, a 3-layer MLP network (of 120-128-64-2 units per layer) with ReLU activations was used. The training configurations is the same as for the 2D maze experiment.The average training time for each environment was about 25 minutes on a Nvidia GeForce RTX 2080 Graphics Card. By applying a linear annealing schedule for the entropy term of the Sinkhorn distance (Eq.(2)), the optimization treats the coupling matrix as a soft assignment problem at the beginning and gradually turns it into a hard-assignment task. In Fig. 4, we show the stability of the optimization of the model parameters by plotting the training Sinkhorn loss for three different environments of the iGibson dataset. In Fig. 5, we show some qualitative results on test trajectories and the model predictions for different iGibson environments.

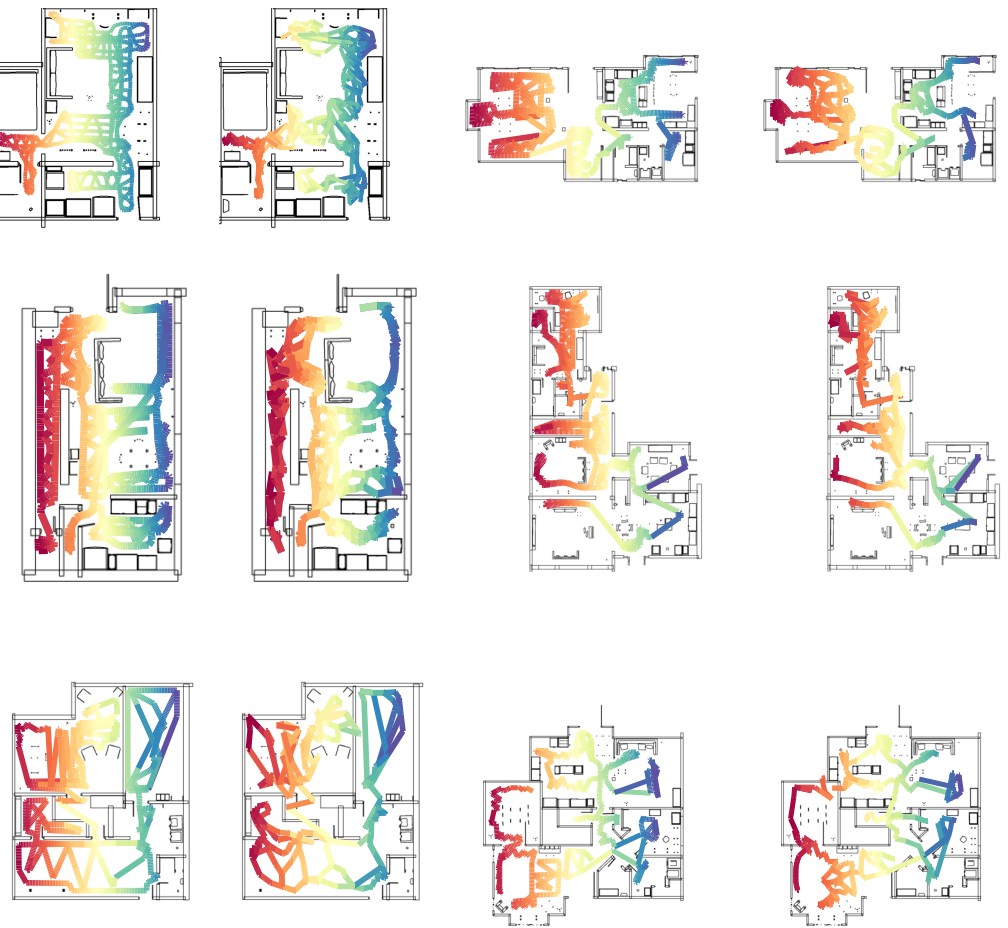

Figure 5: Qualitative results for some iGibson environments; (left) the ground-truth trajectories and (right) the predictions. The colors encode the 2D x-y coordinates of the ground-truth trajectories. For the predictions (right column), we used colors corresponding to those of the ground truth trajectories.

## 4 RF-based localization experiment

We set up an experiment for passive localization of a person using WiFi signals. In WiFi sensing, the superposition of the transmitted signal and all reflected signals from the surface of objects in the environment, including the human body, is captured by the receiver antennas. The WiFi localization at home scale can be a very challenging problem as the signal to noise ratio can be very low as the signal may pass through walls before arriving at the receiver. We collected a data set by using four commercial IEEE 802.11 access points (AP), operating in the 5GHz band, and deployed them in the test environment of size $14 \times 20$ meters in a building. The environment contains three rooms, two long aisles and a large lab (see topological map in Fig. 6). We recorded about 12 minutes of Channel State Information (CSI) of WiFi, while the person traverses in the environment. The WiFi devices communicate about 100 packets per second. So, in total, $72K$ packets are used for training. The same amount of data was used for the test. The ground-truth positions are measured by using multiple 3D cameras, installed in the building. The location of the person's feet on the floor was used as its 2D ground-truth location.

A ResNet-18 is used for spatio-temporal analysis of the WiFi packets and learning the prototype vectors and their KNN samples as explained in Section (3.3.1) of the paper. The CSI tensors are used as the input to the network. Each CSI has dimension of $8 \times 1 \times 208$. The $8 \times 1$ dimensions of each tensor are repeated to have a shape of $8 \times 8$ and then are resized to have shape of $24 \times 24$. This facilitates the application of the standard 2D convolutions on the first two dimensions of the CSI tensor. The third dimension was treated as the channels of the input. The network maps the input

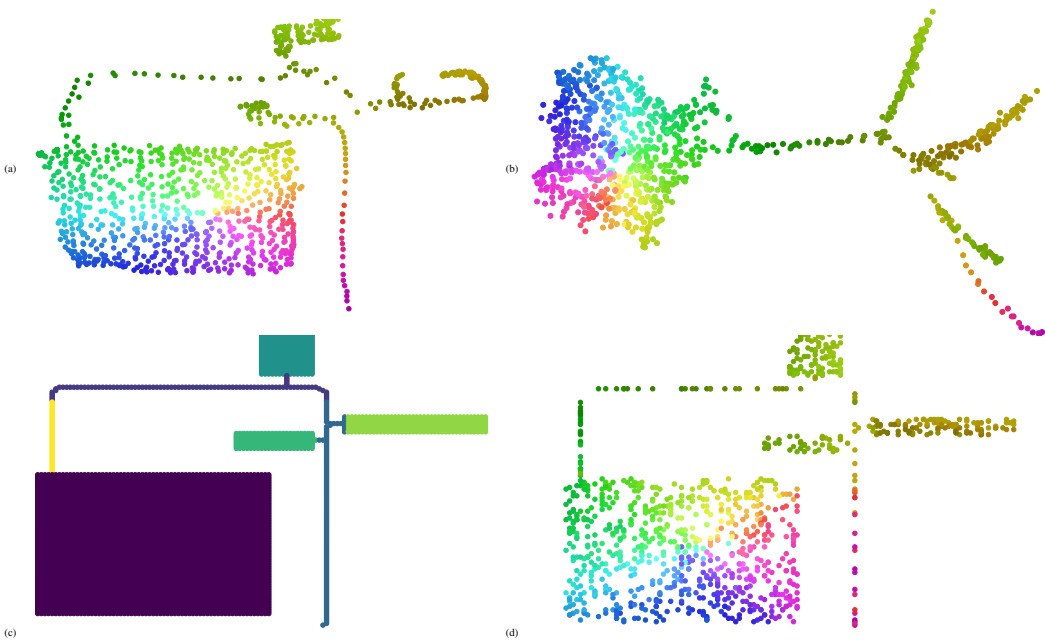

Figure 6: WiFi passive localization; (a) ground-truth positions; colors encode x-y positions of the person; (b) 2D embedding; (c) target topological map with seven zones; (d) predictions.

tensors into a feature space of 6-dimensional. The network is trained by using an Adam optimizer with a learning rate of $1e-4$ for 30 epochs. After training the network, by using the k-means algorithms, the feature vectors are clustered into K=1024 clusters. Consequently, the KNN-graph (K=5) was constructed and all geodesic distances were computed.

We used the same MLP network as the previous two experiments for implementing the map $\Phi$. We used Adam optimizer with a learning rate equal to $1e-3$ and trained the model for 3500 epochs. The range of entropy regularizer for Sinkhorn distance was set as the previous experiments. Fig. 6 visualizes the ground-truth positions of a walking person in the building, the 2D embedding of WiFi packets, the target topological map, and the predictions.