# OpenReview forum: "Modality-Agnostic Topology Aware Localization"
_NeurIPS.cc/2021/Conference — NeurIPS 2021 Poster_

### Official Review · Reviewer_w8Mg · 2021-07-16

**Rating:** 7
**Confidence:** 3

**Summary:**

This paper presents a novel generic localization framework that is agnostic to the data modalities and doesn't require correspondence labels. The paper demonstrates the effectiveness of their framework through experiments on three different environments, each of which has a different data modality.

**Limitations And Societal Impact:**

No potential negative societal impact.

**Main Review:**

Strengths

S1 The idea of designing a generic localization framework is interesting and can inspire future research works.

S2 The design of the proposed framework is novel and clever.

S3 The paper is well written and easy to follow. The theoretical motivation part in Section 3.1 is interesting and inspiring.

S4 The experiments cover three different data modalities, which is great. The experiment results are also very strong.


Comments

C1 I wonder how the proposed framework performs in buildings where there are a lot of similar places. For example, in a hotel, many rooms are identical. If we look at the corridors of a hotel, an apartment, or a dormitory, there could be a lot of visually similar locations.

C2 In Sections 4.2 and 4.3, do we assume that the moving distances between any two consecutive observations are identical and preknown? This seems to be true for the maze environment. I'm not sure if the proposed framework relies on this assumption.


**Time Spent Reviewing:**

5

---

> ### Author Response · Authors · 2021-08-09
> **Answers to the reviewer w8Mg**
>
> We sincerely thank the reviewer for all the positive comments. Below we address the two questions raised.
>
> C1. Visually similar locations easily cause problems for pure image-based location algorithms.  When the similarity between zones/rooms is high, for example between rooms of a hotel, this can even confuse a human. This may also be the case for our proposed model when it is applied on camera inputs, although the triplet sampling and minimizing the marginal loss ensure that the extracted visual features have a higher discrepancy than a marginal constant.  On the other hand, the visual similarity in the environment does not affect the RF-based localization. In the new version of the paper, we would include this discussion.
>
> C2. The proposed model works on temporal sequences of input data. For computational efficiency, we group the measured samples and the center of each group/cluster is assigned to a node in a KNN-graph. Therefore, we do not apply a constant speed constraint on the motion of the observer. While for the maze data a constant speed was used as pointed out by the reviewer, the speed varies for RF-based localization since the person traverses different rooms with U-turns which do not constitute a constant speed.

---

### Official Review · Reviewer_TceL · 2021-07-29

**Rating:** 4
**Confidence:** 4

**Summary:**

This paper presents a new method for localization while being agnostic to input measurement signal's modality. Assuming a map is given, localization inside the map is formulated as a deep manifold learning problem, without needing the supervision of ground truth positions for each input measurement. The method utilizes differentiable optimal transport to solve the correspondence challenge in this formulation. The method is tested on several simulated image datasets and one real-world WiFi dataset.

**Ethical Concerns:**

No.

**Limitations And Societal Impact:**

I like the raw idea of the paper, and the problem being addressed is an important one in robotics. However, the lack of details and the quality of the presentation make it very difficult for me to recommend this paper for acceptance.

**Main Review:**

*Strength*:
1. The problem formulation based on manifold learning seems novel. And the use of optimal transport is an interesting approach.
2. The method allows modal-agnostic localization, due to the use of deep networks (similar to PoseNet).
3. The method is tested on three different types of experiments, showing satisfactory results.

*Weakness*:
Major:
1. Lack of many important technical details:
    1.1. Some important symbols are not explained, which makes it very difficult to understand what has been done. For example, what is $y_j$? And how is $\Omega_t$ sampled (sparely sampled on the trajectory only, or randomly and densely covering the testing environment)? What is V, E in Figure 2?
    1.2. The method did not mention at all how the map $\Phi$ can handle variations in $\Omega_s$ caused by camera rotations, which is a key question on whether the map can localize the camera when it takes two photos at the same position but with different orientations.
    1.3. How are the positive and negative samples in the triplets generated? What's the influence of the hyperparameters in this module (3.3.1) on the whole method?
    1.4. There is no experiment/ablation study to justify the need/importance for equation (6) instead of Euclidean distance in a local area.
2. Lack of discussions of the failure cases. Would this method fail in some cases? For example, the experiments never showed the trajectory in symmetric environments/maps. If $\Omega_t$ is a square or rectangle, how can this method handle the rotational ambiguities? How does it know that an image is taken at the top-left corner instead of the top-right corner (because the optimal transport cost could be the same in this case)?
3. Lack of comparisons to baseline methods, and related work discussions on other weakly supervised training of the localization function $\Phi$ (essentially a PoseNet [Kendall 2015]).
4. Lack of real-world image-based localization experiments.

Minor:
5. Section 3.2 may be shortened since many equations are NOT proposed by this work, but are well-known in the community already. This would save space to include the above missing important details.
6. A topological map has a special meaning in SLAM and localization (no global metric information), which is different from the one used in this paper. It is suggested to change the term to a metric map, because it seems that you do need to make metric measurements (even if it could be noisy) to construct $\Omega_t$.
7. Is it stable to train the deep localization network with the differentiable Sinkhorn? I wish to see some training losses.
8. What if you use L2 instead of KL for L(.,.)? Any ablation study? Do you have any reference to support your claim that "choosing Eucidean distances for L is ill-posed"?

**Time Spent Reviewing:**

4 hours

---

> ### Author Response · Authors · 2021-08-09
> **Answers to the reviewer TceL**
>
> We sincerely thank the reviewer for all the constructive comments. Below we address the questions raised.
>
> **1.1** In Algorithm 1 at line 191, the set $Y \in  \Omega_t$ was introduced as the target sample set. The symbol $y_i$ indicates a member of $Y$. In Figure (2), the two symbols V and E denote the nodes and edges (i.e. KNN indices) of the KNN-graph, respectively. We would add the missing definitions to a final version.
>
> Additionally, we will consider explaining that the sampling from target map was done randomly and densely covers the testing environment. The number of target samples is higher than the number of prototype vectors. Using the optimal transportation, only a subset of the target samples may be assigned to the sampled input data, based on the cost of transportation.
>
> **1.2** In line 260, we mentioned that four orthogonal-view images are used for constructing a panorama image. These four images are created by rotating the camera around the up-axis in steps of 90 degrees. In new version of paper, we will explain that for handling the variations caused by camera rotation, we know that cyclic rotations of panorama images are equivalent to an addition of linear phase in the Fourier domain [1], [2]. Thus, we first compute the FFT of each panorama image and then by estimating and removing the linear phase in the horizontal direction and transforming it back into image domain by using the inverse FFT, we normalize each panorama image. Although using a rotation-invariant network (e.g. [3]) can be considered as an alternative solution, such solutions strongly depend on the input modality and do not generalize to other modalities.
>
> 1. T. Pajdla and V. Hlavac, "Zero phase representation of panoramic images for image based localization". In LNCS, vol. 1689, Springer, 1999, pp. 550–557.
>
> 2. Stürzl, Wolfgang and Hanspeter A. Mallot. "Efficient visual homing based on Fourier transformed panoramic images." Robotics and Autonomous Systems 54(4), 2006, pp. 300-313.
>
> 3. B. Chidester, M. Do and J. Ma. "Rotation equivariance and invariance in convolutional neural networks." arXiv preprint arXiv:1805.12301 (2018).
>
> **1.3** Triplet samples are chosen based on their temporal vicinity (see line 212 of the paper). Since the input to the model is assumed to be a temporal sequence(s), in each sequence, for each sample (anchor), we consider one small and one large temporal window with predefined fixed widths. These two temporal windows are centered at the anchor. Any sample inside the smaller temporal window can be considered as a positive sample and any sample outside the small window but inside the large window can be considered as a negative sample.
>
> Regarding the choice of hyper-parameters in section 3.3.1, below we explain separately the upper bound of selecting negative samples, the number of clusters $N_s$ and the neighbor size $K$ in the KNN computation.
>
> The widths of the temporal windows roughly depend on the speed of the observer in the environment. The idea is that the negative samples needs to include data at which the observer has sufficient displacement from its previous location (i.e. anchor sample). The upper bound of the width of the larger temporal window is set as large as possible. However, if it is chosen too large, the observer could revisit the same location. In other words, the observer returns to the location where the anchor sample was collected.
> The number of clusters $N_s$ is a trade off between spatial accuracy and computational cost. A larger number of clusters leads to a denser sampling from the data manifold but it requires training the model on a larger set to learn the embedding and to find the transportation cost to the target domain.
> The neighborhood size $K$ should be large enough to guarantee the connectivity in the KNN-graph but too large values degrade the expressiveness of KNN-graph and the accuracy of its computed geodesic distances. We will add these explanations to the supplementary material.
>
> **1.4** In line 225, we cited the work of Dsilva et al. 2015, who address the importance of employing the Mahalanobis distance instead of the Euclidean distance. Based on their work, using the Mahalanobis distance (Eq. (6)) is representative of distances along the local principal direction of data manifold. A comparison with the Euclidean distance was given in their work (please see Figure 4 in their paper). We will more prominently address this point in our explanation in the paper.
>
> **2** While we did not have fully symmetric environments in our datasets, we can confirm that symmetries cause ambiguities. For instance, if the environment contains only a single room/zone with symmetrical shape, the obtained solution is valid up to an isometric transformation. In such a special case, the final solution requires a few landmarks (for instance 3 points in 2D space) for computing the isometric transformation. We will consider adding a similar discussion in the new version of the paper.
>
> **3** Posenet [Kendall 2015] as well as newer works such as
> * Learning Less is More – 6D Camera Localization via 3D Surface Regression by Brachmann and Rother, CVPR 2018
> * Back to the Feature: Learning Robust Camera Localization from Pixels to Pose, by Sarlin, Germain, Toft, Larsson, Pollefeys, Lepetit, Hammarstrand, Kahl and Sattler, CVPR 2021
> * Coming Down to Earth: Satellite-to-Street View Synthesis for Geo-Localization by Toker, Zhou, Maximov and Leal-Taixe, CVPR 2021
>
> require strong supervision like known position and orientation (PoseNet), full 3D models (Learning Less is more and Back to the Feature), or image pairs (Coming Down to Earth). Furthermore, these algorithms work for very specific data only.
> In contrast, the proposed algorithm can be trained without those supervisions and it can be applied to very different modalities. Up to the best knowledge of the authors, there is no localization algorithm which works on multiple modalities like the proposed one.
>
> **4** We agree that our image-based experiment is on simulated data, and would like to draw attention to our real-world experiment using the WiFi modality. We would like to emphasize that this is not a contrived experiment to just broaden scope but an important signal that has recently garnered a lot of attention e.g. Making the Invisible Visible: Action Recognition Through Walls and Occlusions (ICCV 2019), Through-Wall Human Pose Estimation Using Radio Signals (CVPR 2018), etc. Positioning through WiFi can be useful in the domain of home, enterprise and industrial automation and robotics. Examples include intrusion detection, asset tracking, smart energy usage. It is not affected by poor light conditions and can thus work in the dark, and it can work across walls. Also, a device without a video-camera is more likely to be adopted by a user since it largely mitigates issues of privacy related to video images. Our experimental setup is designed to mimic a real-world deployment and thus we only use three receivers and one transmitter for a large multi-room environment where most areas lack line-of-sight.
>
> **5** We will consider the reviewer comments by shortening section 3.2 and referring the reader to the previous works.
>
> **6** We thank the reviewer for the suggestion and would change it in the final version.
>
> **7** In line 187 of the paper, we explained and referred to other works that the entropy regularization of the Sinkhorn distance makes the optimization stable. By applying a linear annealing schedule for the entropy term, the optimization treats the coupling matrix as a soft assignment problem at the beginning and gradually turns it into a hard-assignment task. We will add the visualization of the training loss to the supplementary materials.
>
> **8** In Eq. (1), the term $L(D_s, D_h)$ compares two distance matrices. The matrix $D_s$ contains pairwise geodesic distances along tangent planes in the ambient space of the signal and $D_h$ measures the pairwise Euclidean distances in the latent space. As discussed in Cuturi et al. 2013, choosing the $L2$ distance between two distance matrices leads to a measure-preserving isometry while we know that the geodesic distances are equal to the Euclidean distances only in a convex region so preserving the isometry is not required.

---

### Official Review · Reviewer_ynTh · 2021-08-03

**Rating:** 8
**Confidence:** 3

**Summary:**

The reviewed paper considers the problem of 2D localization of an observer on a given topological map. A data-driven approach is proposed to solve the problem. While the topological map of the environment is required, no prior information is needed about the nature of the input signal. Moreover, no supervision (in terms of pairs of signal measurement and observer's location) is needed.

The work formulates the localization problem as the problem of learning a low-dimensional manifold, representing the input in their intrinsic space and then transport them to the given topological map by inferring correspondences between the two spaces.
The work is built on two intuitions:
1. that the transformation from high-dimensional input to the intrinsic space can be modeled as a mapping that can be learned through a neural network;
2. temporal data is subject to gradual changes as the observer moves, suggesting that the data reside on a smooth, locally connected manifold;

Results from experiments on three different environments (and different sensory inputs) support the claims.


**Limitations And Societal Impact:**

The authors should consider a more extensive description of the limitations of the proposed approach. For example, in the experiments reported in the document, the only source of disturbance/signal was the observer and its motion itself. What happens when other people/objects occupy/walk the environment at the same time? Obviously, in these scenarios, solutions to localization involving cameras have a clear advantage, but it would be useful to the reader to get a sense of what kind of problems can arise in those scenarios.

No clear presence of potential negative social impact in this work.

**Main Review:**

The proposed solution to the 2D localization problem appears to be technically sound and well documented.
The document is clearly written and organized.
Though the amount of information and references is large, the document leaves a reader familiar with the subject with enough pointers to reproduce the work.

A few minor typos:
- Line 162: "Euc**l**idean"
- Line 186: "Re**q**ularizing"
- Line 252: Missing reference to a figure is shown as "??"

**Time Spent Reviewing:**

2

---

> ### Author Response · Authors · 2021-08-09
> **Answers to the reviewer ynTh**
>
> We sincerely thank the Reviewer for all the positive comments. Below we describe the limitations of the proposed approach that would be considered in the new version of the paper.
>
> In our proposed method, we assume that the main source of variations in data is due to the displacement of the agent/person in the environment. Motion of other objects/people in the environment can affect finding the precise KNN samples. However, since we use a contrastive learning for grouping the samples in spatial domain, based on their timestamp, the network can discard the temporal irrelevant information in the scene. We would like to further point out our ablation study (section 4.4) in which we demonstrated the robustness of the model against variations in object appearance and texture in the scene.
>
> As mentioned by the reviewer, this problem is less important for the camera-based localization and in general for an active localization problem (i.e. when the observer carries a device which is being used for measurement like RF signal of a cell-phone or a camera).
> In passive RF-based localization case (i.e. when only the reflected signal from the surface of observer body is the source of information), motions of other people cause problems as then the measured signal contains a superposition of more than one sample on the data manifold. Thus, in the passive scenario, the method requires sampling from the data manifold by a single observer but can handle stochasticity, interference signals and noise.

---

### Decision · Program_Chairs · 2021-09-27

**Decision:**

Accept (Poster)

**Comment:**

The paper proposes an algorithm for 2D topological localization that is largely agnostic to the type of observation data. Formulated as a deep manifold learning problem, the framework transforms the (potentially) high-dimensional observation into a low-dimensional embedding. The method then uses optimal transport to identify the correspondence between this representation and that of the topological map. Importantly, this correspondence is learned without ground-truth correspondences. The paper presents experimental results using simulated camera observations as well as radio frequency (WiFi) observations.

Localization has been researched in the robotics community for several decades and is considered by many to be a problem that is all but solved. As the paper notes, traditional methods employ measurement models that are specific to the particular sensing modality(ies). To some extent, these models provide an abstraction of the specific sensor, and the underlying localization algorithms tend to be more sensor-agnostic. The capacity for the end-to-end framework to be truly independent of the particular sensor, however, is compelling. The way in which the method employs manifold learning and optimal transport is interesting and is technically sound. The experiments demonstrate the algorithm's ability to generalize to very different sensing modalities, however experiments involving real rather than simulated images would provide more compelling evidence. The reviewers raised a few concerns regarding the need for a clearer discussion of the algorithm's limitations and the omission of various technical details, which the authors largely clarified in their response. These issues should be addressed and the authors are encouraged to consider the possibility of including an experimental evaluation on real-world imagery.